# Monitoring and Management of Cytomegalovirus Reactivations After Allogeneic Hematopoietic Stem Cell Transplantation in Children: Experience from a Single Pediatric Center

**DOI:** 10.3390/diagnostics14212461

**Published:** 2024-11-03

**Authors:** Giulia Ferrando, Francesca Bagnasco, Stefano Giardino, Filomena Pierri, Sara Pestarino, Eddi Di Marco, Maria Santaniello, Elio Castagnola, Maura Faraci

**Affiliations:** 1Infectious Diseases Unit, Department of Pediatrics, IRCCS Istituto G. Gaslini, 16147 Genova, Italy; giuliaferrando@gaslini.org (G.F.); mariasantaniello@gaslini.org (M.S.); eliocastagnola@gaslini.org (E.C.); 2Epidemiology and Biostatistics, Scientific Directorate, IRCCS Istituto G. Gaslini, 16147 Genova, Italy; francescabagnasco@gaslini.org; 3Hematopoietic Stem Cell Transplant Unit, Department of Hemato-Oncology, IRCCS Istituto G. Gaslini, 16147 Genova, Italy; filomenapierri@gaslini.org (F.P.); sarapestarino@gaslini.org (S.P.); maurafaraci@gaslini.org (M.F.); 4Molecular Medicine Laboratory, IRCCS Istituto G. Gaslini, 16147 Genova, Italy; eddidimarco@gaslini.org

**Keywords:** hematopoietic stem cell transplantation, viral infection, cytomegalovirus, pre-emptive therapy

## Abstract

Background: CMV reactivation represents a frequent complication after HSCT. The aim of this study was to describe the incidence of CMV reactivation in a pediatric HSCT cohort and analyze the potential impact of recipient/donor-related or transplant-related factors on this complication. Furthermore, we analyzed the management of CMV reactivation in order to purpose criteria for pre-emptive therapy. Methods: Allogeneic HSCTs, performed at IRCCS Istituto Gaslini between 2012 and 2022, were included in this analysis. CMV–DNAemia was regularly monitored. Risk stratification was based on donor/recipient serological status and additional potential risk factors were considered: haploidentical transplant; any HSCT subsequent to the first; acute and chronic GvHD; steroids; and other immunosuppressive therapies. We described also the approach for pre-emptive therapy during the period 2012–2019. Results: A total of 214 allogeneic HSCTs were performed in 189 patients. In total, 100 (46.7%) HSCTs were complicated by at least one reactivation. CMV reactivation was significantly associated with high serological risk and steroid treatment. Pre-emptive therapy was administered in 59/69 (85.5%) HSCTs during 2012–2019. In the presence of predefined risk conditions, therapy was started at a median viremia of 2050 copies/mL. No difference was observed in OS between patients with CMV reactivation versus patients who did not present this complication. Conclusions: These results suggest the potential effectiveness of the approach used in providing pre-emptive therapy based on viral load monitoring and individualized risk factors.

## 1. Introduction

Cytomegalovirus (CMV) reactivation represents the most frequent viral complication after allogeneic hematopoietic stem cell transplantation (HSCT), occurring in about 70% of CMV seropositive HSCT recipients. Recurrence rates after a first CMV–DNAemia can be as high as 50–70% [1]. The incidence of CMV reactivation is higher during the first 100 days after HSCT (early-onset CMV disease) when CMV-specific immunity has not yet recovered. Early CMV reactivation is associated with poor outcomes, and the overall mortality among these patients remains high [2]. This is due to the negative direct and indirect impact of CMV, as well as an increased risk for secondary bacterial and fungal infections and GvHD [3]. Therefore, during this period, it is strongly recommended to perform strict weekly CMV surveillance of viral DNAemia by quantitative polymerase chain reaction (qPCR) [4,5]. Depending on the laboratory, two units of measure can be used to monitor viremia: copies/mL or the WHO standard IU/mL (International Units, IU). An example of a conversion factor from copies to IU is 4 copies = 1 IU [6]. However, it is important to note that there is not a single conversion factor because laboratories use different kits, which can lead to variability in measurements.

Since CMV reactivation may also occur later, the monitoring of CMV–DNAemia should be continued after day +100 according to the clinical condition and immunological status of patients [2,5,7]. This includes monitoring in the presence of at least one of the following conditions: lymphopenia (<100 lymphocytes/mm^3^); positive CMV–DNAemia before day 100; Graft-versus-Host Disease (GvHD), requiring a methylprednisolone dose > 0.5 mg/kg/day; and the absence of CMV-specific T-cell immunity [8].

While CMV reactivation is frequent, CMV disease is rare, with its incidence ranging from 5% to 25% [9,10]. It may involve different organs, causing pneumonitis, hepatitis, retinitis, gastroenteritis, and encephalitis [10].

CMV viremia is a well-recognized risk factor for CMV disease and has led to a strategy of early administration of antivirals triggered by predefined CMV-DNA levels (pre-emptive therapy). These levels are defined on the blood compartment analyzed (plasma or whole blood) and the quantification method used. In these settings, viral load reduction is currently used to monitor the response to treatment [11].

Ganciclovir, valganciclovir, and foscarnet are commonly used as monotherapies for pre-emptive therapy, while their combination in case of further and/or non-responding reactivation is debated [8]. However, their use is potentially associated with adverse events that can sometimes be significant, such as cytopenia (ganciclovir) or kidney function impairment (foscarnet). To the authors’ knowledge, there is no shared validated quantitative CMV–DNAemia threshold for initiating pre-emptive therapy in the pediatric HSCT setting. In adults affected by CMV reactivation, the viral threshold to start a pre-emptive therapy is proposed based on patients’ stratification into two risk groups [2]. The “high risk” group includes HSCTs from HLA-mismatched or haploidentical donors, from cord blood as a stem cell source, and transplanted patients previously treated with a prolonged course of steroids (>1 mg/kg/day of prednisone or equivalent for at least one month). In this group, the reference criteria for initiating pre-emptive therapy are either CMV-DNA ≥ 150 IU/mL or DNA levels rising more than five times above baseline within one month during the first 100 days after HSCT in patients already receiving letermovir as prophylaxis. The threshold is lowered to 50 IU/mL in patients with no ongoing prophylaxis. The “low risk” group includes all patients not fulfilling the high-risk criteria. In this category, pre-emptive therapy is started with CMV-DNA ≥ 150 IU/mL during the first 100 days after HSCT if the patient is not receiving any prophylaxis; or ≥500 IU/mL or in case of the DNA level increasing to more than five times the baseline, in patients already receiving letermovir prophylaxis. After the first 100 days, the threshold is always ≥500 IU/mL regardless of prophylaxis and risk category.

This retrospective monocentric study aimed to investigate the incidence of CMV reactivations after allogenic HSCT and analyzing the potential impact of recipient/donor-related and transplant-related factors in pediatric patients. The secondary objective was to describe and analyze the management of pediatric patients with CMV reactivation to propose criteria for starting pre-emptive therapy.

## 2. Materials and Methods

### 2.1. Study Design

This is a retrospective single-center study conducted at IRCCS Istituto G. Gaslini in Genoa (Italy) and approved by the Institutional Ethics Committee on 11 November 2019 (number 273/4668). All allogeneic HSCTs performed in patients aged ≤ 25 years between 2012 and 2022 at the HSCT Unit of Istituto G. Gaslini, with follow-up censored in July 2023, were included. Written informed consent was obtained from the patients’ parents.

Data on CMV reactivation were extracted from the database of the Laboratory of Molecular Biology and analyzed retrospectively. Demographic and HSCT features and outcomes were obtained from the clinical records. Data on the use of pre-emptive therapy were collected for the period 2012–2019.

Quantitation of CMV copies in whole blood was performed using real-time polymerase chain reaction (CMV ELITe MGB kit from ELITech Group SpA, Turin, Italy), considering that the sensitivity limit of the technique is 250 copies/mL and the upper limit of the technique is 10^7^ copies/mL. CMV–DNAemia was monitored twice a week by polymerase chain reaction (PCR) on blood samples, from the start of the conditioning regimen until day +100 after HSCT, regardless of the serological status of donor and recipient, or for a longer period in patients with delayed immune recovery. Episodes of CMV reactivation before these periods were excluded.

HSCTs were classified according to donor type as “MRD” for matched related donors (HLA genotypically identical), “haplo” for haploidentical related donors, and “AD” for alternative HLA-matched unrelated donors from any source. In the case of patients who received more than one HSCT, each procedure was considered separately. The stem cell sources were bone marrow (BM), peripheral blood (PBSC), or cord blood (CB). The conditioning regimen (CR) was categorized as myeloablative (MAC) or reduced-intensity (RIC) [12].

Neutrophil and platelet engraftment were defined as the first of 3 consecutive days of achieving a sustained peripheral blood neutrophil count of >0.5 × 10^9^/L and as the first of at least 7 days with a platelet count of more than >20 × 10^9^/L without any transfusion during the 5 days prior, respectively.

Acute (a-) GvHD and chronic (c-) GvHD were classified according to the NIH criteria [13]. All patients received acyclovir as herpes virus 1–2 prophylaxis (30 mg/kg/day IV or 80 mg/kg/day oral administration) after HSCT until day +100 in the presence of adequate immune reconstitution.

Concerning donor/recipient CMV serological status evaluated before HSCT, patients were defined as (1) low-risk in the presence of negative CMV serology in both recipient and donor (R−/D−); (2) standard-risk with positive CMV serology in the donor regardless of recipient status (R+ or −/D+); or (3) high-risk with negative CMV serology in the donor and positive in the recipient (R+/D−).

In addition, the following factors were considered as predefined conditions at risk for CMV reactivation, as previously described [8]:Haploidentical transplant;Any HSCT subsequent to the first;aGvHD;cGvHD;Steroid dose > 2 mg/kg/day;Second-line immunosuppressive drugs for the treatment of GvHD.

### 2.2. Statistical Analysis

Descriptive statistics were reported in terms of absolute frequencies and percentages for categorical data, and the Pearson’s chi-square test or Fisher’s exact test, if appropriate, were applied to compare proportions. Continuous data were described in terms of median values, range, and first and third quartiles due to their non-normal (Gaussian) distribution. Accordingly, comparisons between groups were made by the non-parametric Mann–Whitney U-test.

The counting process approach was applied to consider that any patient could have received more than one HSCT.

The association between the binary outcome variable (CMV reactivation) and independent variables (demographic and HSCT features) was assessed by a standard logistic regression model and reported in terms of odds ratio (OR) and 95% confidence interval (CI) with the robust estimator of variance allowing for intra-group (intra-patient) correlation. The likelihood-ratio test (LR) was used to measure the effect of each predictor. Variables that were significantly associated with the study outcome were identified using a backward selection procedure.

To adjust the analysis for competing risks, the CMV reactivation curve after HSCT was estimated by the crude cumulative incidence method, with death as the competing risk.

In the group of patients with or without CMV, the cumulative probability of survival since HSCT was estimated by the Kaplan–Meier method and was expressed as a percentage with 95% CI. The Log-rank test was applied to compare the cumulative survival between groups with and without CMV; the CMV variable was modeled as a binary time-dependent covariate since HSCT.

All tests were two-tailed and a *p*-value < 0.05 was considered statistically significant. All analyses were performed using Stata (StataCorp. Stata Statistical Software, Release 18.0 College Station, TX, USA, Stata Corporation, 2023).

## 3. Results

### 3.1. Characteristics of 214 Allogeneic HCTs

During the study period, 214 allogeneic HSCTs were performed in 189 patients with both malignant and non-malignant diseases. Among them, in four patients, we analyzed the second HSCT because they received the first transplant before the start of the study. Moreover, 17 patients received two HSCTs and 4 children received three HSCTs during the period of our analysis, for a total of twenty-nine HSCTs subsequent to the first. The median age at transplant was 7.9 years (1st–3rd quartiles 3.3–13.6). The donor was an AD in 86 (40.2%), a haplo in 85 (39.7%), and an MRD in the remaining 43 (20.1%) transplants. Among haplo-HSCTs, 46 (54.1%) were performed with ex vivo TCRαβ/CD19 depletion, and 39 (45.9%) were T-replete haplo with unmanipulated bone marrow and post-transplant cyclophosphamide as GvHD prophylaxis. Bone marrow was the graft source in the majority of cases (65.4%). Engraftment was reached after a median of 17 days (1st–3rd quartiles 14–22) in 198 HSCTs and of 21 days (1st–3rd quartiles 14–28) in 179 HSCTs, respectively, for neutrophils and platelets, while 28 HSCTs were complicated by graft failure, 11 of which were secondary.

### 3.2. CMV Reactivations and Risk-Factor Analysis

In this cohort, 100/214 (46.7%) HSCTs were complicated by at least one CMV reactivation, 49/100 (49%) by ≥2 episodes, and 20/100 (20%) by ≥3 episodes. The cumulative CMV incidence at 100 days from HSCT was 44.5%, 95% CI (37.3–51.6). The median time to the first CMV reactivation was 22 (1st–3rd quartiles 12–41, maximum 393) days after HSCT.

The analysis of the risk factors reported in Table 1 indicates that CMV reactivation was significantly associated with the presence of high serological risk (D−, R+) and high-dose steroid treatment (MPD > 2 mg/kg/day). These results were confirmed also by the logistic regression model; HSCTs with high serological risk had a two-fold risk of CMV reactivation compared to HSCTs with low/intermediate serological risk, OR = 2.0, 95% CI (1.0–3.9), *p* = 0.038. For high-dose steroid treatment, the OR was 2.5, 95% CI (1.3–5.0), *p* = 0.008.

Among the 100 CMV reactivations, only 1 was associated with a CMV end-organ disease (retinitis). This complication occurred three months after an AD HSCT performed for Fanconi anemia in a patient with high serological risk status.

At the last visit (median follow-up of 3.1 [1st–3rd quartile 1.2–5.5] years), a total of 48 deaths (25.4%) were reported among 189 patients, with 23 (24.5%) deaths occurring in 94 patients with CMV reactivation. It is noteworthy that no deaths directly related to CMV were observed, as shown in Table 1. Considering the time dependence since HSCT of CMV reactivation, a better cumulative survival was observed among patients without a reactivation compared to those with a reactivation, although not reaching statistical significance, *p* = 0.2542. The 5-year cumulative survival was 76.7%, 95% CI (66.5–84.2), and 69.1, 95% CI (56.8–78.5), respectively, for patients without and with a CMV event, as shown in Figure 1.

### 3.3. Pre-Emptive Therapy

Among the 163 allo-HSCTs performed in the period 2012–2019, 69 (42.3%) were complicated by at least one CMV reactivation, with 34 (20.6%) experiencing ≥2 episodes and 15 (9.2%) having ≥3 episodes. Nine (13%) CMV reactivations were reported after a transplant subsequent to the first. The median time at first CMV reactivation was 23 days (1st–3rd quartiles 15–43 days) from HSCT.

Pre-emptive therapy was administered in 59/69 (85.5%) HSCTs and there was no statistically significant association between the HSCT characteristics and the initiation of pre-emptive therapy (Table 2). The only significant differences between the treated and untreated group were having at least one predefined risk condition and a higher number of viral copies. Among the 59 HCTs in which pre-emptive therapy was administered, 52 (88.1%) had at least one of the predefined conditions considered at risk for CMV reactivation/disease, and the pre-emptive therapy was started at the median viremia of 2050 copies/mL (1st–3rd quartiles 800–4150). In the other seven cases without any predefined conditions, the pre-emptive therapy was initiated in the presence of a higher level of CMV DNA (median viremia of 5700 copies/mL, 1st–3rd quartiles, 400–9400). Conversely, in four patients, pre-emptive therapy was started in the presence of a lower viral load of 200 copies/mL because they developed CMV reactivation in the pre-transplant or pre-engraftment phase. For all the latter four cases, early reactivation was considered as an individual risk factor.

Out of the 10 cases where pre-emptive therapy was not administered, 4 (40%) had predefined risk conditions but did not receive the therapy because a low viral load (max 1650 copies/mL in one determination) was observed for a brief period.

Ganciclovir or valganciclovir usually constituted pre-emptive therapy, while foscarnet was administered as a second-line therapy in case of failure with ganciclovir or as a first option in case of CMV reactivation during the pre-engraftment period. Cell therapy with antiviral CMV-specific T-cells was employed in patients lacking a response or experiencing progression after at least two lines of pharmacological therapies and/or in those with severe antiviral drug-related toxicity. Five patients received CMV-specific T-cells for these reasons, one received three infusions from a third-party donor, and the remaining four received one infusion from the same stem cell donor. Except for a mild inflammatory syndrome in one case, which rapidly resolved without intervention, all infusions were well tolerated, and all patients achieved a complete response. One patient experienced another CMV reactivation after antiviral T-cell infusion, which responded promptly to ganciclovir.

## 4. Discussion

CMV reactivation remains one of the most frequent complications after allogeneic HSCT, with a historically higher incidence in adults than in the pediatric population, related to the higher prevalence of positive CMV serology and to the slower post-HSCT immune recovery in adults [8,14]. However, the recent introduction for adult HSCT recipients of effective CMV prophylaxis with letermovir, not yet registered in children, is destined to change the history of CMV-related morbidity in this population. Conversely, the increased frequency of haploidentical transplantation with in vivo or ex vivo T-cell depletion in the pediatric population might increase the incidence of CMV in children as well, making CMV monitoring and a better-defined pre-emptive therapy approach even more crucial.

The reported incidence of CMV reactivation of 46.7% in our cohort, which includes HSCT from haploidentical donors and second transplants potentially at higher risk, aligns with the range reported in the pediatric literature [5].

The donor–recipient serostatus (seronegative donor/seropositive recipient) has been strongly confirmed as having the greatest impact on the risk for CMV reactivation. This point is crucial since it defines recipients at higher risk for CMV-related complications, but it may also guide, when possible, donor choice [15].

Furthermore, our analysis showed a direct impact of post-transplant immunosuppressive treatment on CMV reactivation, with high-dose steroid treatment, usually used as a first-line treatment for both acute and chronic GvHD, as the second main risk factor. This confirms the role of delayed immune reconstitution in CMV reactivation, which is more common in recipients who develop GvHD, requiring prolonged steroid treatment. This indirectly highlights the role of GvHD in CMV reactivation, although the limited number of observations in our population did not enable us to show its significant impact.

Moreover, among patients who experienced CMV reactivation, 11 developed aGvHD only after the first episode of reactivation, but all of them presented a subsequent reactivation. This confirms, in our cohort, the role of CMV in favoring the development of GvHD, the treatment of which, in turn, may lead to further viral reactivations.

The bidirectional relationship between acute GvHD and CMV has been extensively described [16,17,18] but the underlying causative factors remain speculative. CMV-specific T-cells’ cross-reactivity versus host allo-antigens and the CMV-related imbalance in T-reg recovery have been claimed to explain the increased GvHD in HSCT recipients who experience CMV reactivation. Likewise, the impairment of donor-derived alloreactive T-cells could limit the immune control of CMV. Conversely, second-line treatments such as Etanercept and/or extracorporeal photochemotherapy, used in steroid-dependent or refractory GvHD, did not seem to add an additional risk for CMV reactivation in this analysis. Similarly, no significant increase in risk emerged from the use of Jak-inhibitors or other tyrosine kinase inhibitors for chronic GvHD. This could be attributed to extracorporeal photochemotherapy being the “second-line treatment” in a high number of patients, resulting in a lower immunosuppression status than other second-line therapies.

These findings could support monitoring plans and challenging clinical decisions about the indication for pre-emptive therapy in pediatric HSCT recipients that should be individualized and patient-centered.

Based on the results of the risk factor analysis between HSCT recipients with and without CMV reactivation and some previous literature suggestions, we defined risk factors and categories as shown in Figure 2. Donor–recipient serostatus and the presence of high-dose steroid treatment for GvHD play a crucial role, while although a significant association between “haploidentical or mismatched donor”, “subsequent HSCTs”, and CMV reactivation has not been found clearly in our analysis, we consider also these conditions to be at “risk”, because of the potential slower immune reconstitution observed in these cases. The low number of subsequent HSCTs (*n* = 29) may explain the failure to achieve statistical significance for these risk factors. Of note, the first CMV reactivation occurred most frequently in the first weeks after the transplantation, at a median of 23 days, showing the early post-transplant phases to be the periods at higher risk. In the four patients who developed positive CMV viral load in the pre-transplant or pre-engraftment phase, pre-emptive therapy was started in the presence of a viremia ≥ 200 copies/mL, like the threshold values suggested for adults [2], considering early viral reactivation as an individual risk factor. We suggest considering this as a “timing-related risk factor” that justifies the early start of pre-emptive therapy.

In our study, we also described the approach followed for starting pre-emptive therapy during the period from 2012 to 2019. About 40% of allogeneic HSCTs had a CMV reactivation and most of them (*n* = 59, 85%) received treatment. Although nearly half of them experienced more than one episode, only one case of CMV end-organ disease was reported and none of our patients died as a consequence of CMV reactivation. Moreover, no significant difference was observed in OS between those who showed reactivated CMV versus those who did not. These results could suggest the effectiveness of the approach used in this cohort in providing pre-emptive therapy, started in the presence of different viral load thresholds based on individual and timing-related risk factors.

While the main limitation of this study is its retrospective and single-center nature, it provides an opportunity to address important questions regarding the individual risk factor-oriented approach for starting pre-emptive CMV treatment in the pediatric population, enabling us to derive useful information for this challenging management.

However, as previously defined [16], the challenge is to validate an evidence-based CMV viral load threshold for starting pre-emptive therapy. We, therefore, propose a risk stratification for the management of CMV reactivations in pediatric patients (Figure 2), adapting the recommendations for adult populations [2] to our results in children, although further larger and prospective studies are required to validate its effectiveness to prevent CMV disease and also to avoid overtreatments and consequent side effects from the use of antiviral drugs. Considering the rate of recurrent CMV reactivation in our cohort, reduced efficacy in the presence of delayed immune reconstitution of anti-CMV drugs, and hypothetical CMV antiviral resistance, represent the major issues to overcome [7,17]. A cell-therapy approach with virus-specific T-cell therapy (from stem cell donors or third parties) should be considered in cases of persistent or recurrent CMV reactivation, although currently, the technologies and expertise required for these approaches are limited to a few transplant centers [15,19]. Another useful tool to improve the treatment of CMV reactivation could be the study of CMV-DNA mutations, although it is currently far from moving from research to clinical practice. For these reasons, optimizing an effective approach with viral load monitoring and starting pre-emptive treatment based on individualized risk factors appear crucial. This report could be a useful start to plan ad hoc prospective pediatric studies.

## Figures and Tables

**Figure 1 diagnostics-14-02461-f001:**
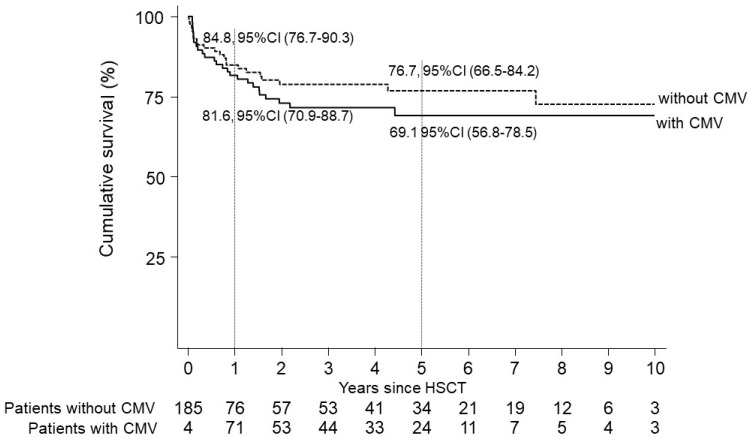
Cumulative survival in patients with or without CMV reactivation.

**Figure 2 diagnostics-14-02461-f002:**
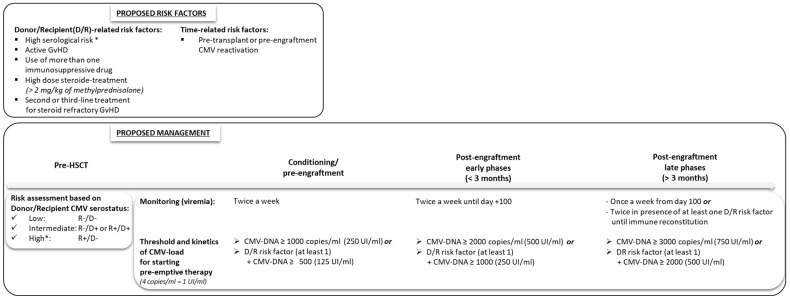
The proposed risk stratification and management in pediatric HSCT recipients.

**Table 1 diagnostics-14-02461-t001:** Analysis of potential risk factors for CMV reactivations.

Factors	Total, *n* = 214	No Reactivation, *n* = 114	Reactivation, *n* = 100	*p*-Value
Age at HSCT, years, median (1st–3rd quartiles)	7.9 (3.3–13.6)	7.8 (2.8–13.6)	8.0 (3.4–13.7)	0.536
Sex, Males, *n* (%)	131 (61.2)	70 (61.4)	61 (61.0)	0.952
Non-malignant disease, *n* (%)	111 (51.9)	60 (52.6)	51 (51.0)	0.812
High serological risk, *n* (%)	46 (21.5)	18 (15.8)	28 (28.0)	**0.030**
Subsequent HSCT, *n* (%)	29 (13.5)	14 (12.3)	15 (15.0)	0.562
Donor type, *n* (%)				0.169
Haplo	85 (39.7)	52 (45.61)	33 (33.0)	0.069
*TCR-αβ^+^/CD19^+^-depleted*	46 (54.1)	30 (57.7)	16 (48.5)	0.406
*PT-Cy*	39 (45.9)	22 (42.3)	17 (51.5)	
AD	86 (40.2)	41 (36.0)	45 (45.0)	
MRD	43 (20.1)	21 (18.4)	22 (22.0)	
Second-line immunosuppressive therapy, *n* (%)	41 (19.2)	19 (16.7)	22 (22.0)	0.385
Methylprednisolone > 2 mg/kg, *n* (%)	43 (20.1)	15 (13.2)	28 (28.0)	**0.007**
Acute GvHD, *n* (%)	96/195 (49.2)	48/100 * (48.0)	48/95 * (50.5)	0.724
Acute GvHD grade III-IV, *n* (%)	55/195 (28.2)	23/100 * (23.0)	32/95 * (33.7)	0.063
Chronic GvHD, *n* (%)	46/182 (25.3)	18/91 * (19.8)	28/91 * (30.8)	0.088
Chronic GvHD extensive, *n* (%)	20/182 (11)	7/91 * (7.7)	13/91 * (14.3)	0.615
Presence of at least one predefined condition at risk for CMV **	174 (81.3)	93 (81.6)	81 (81)	0.914
Deaths, *n* (%)	48/189 *** (25.4)	25/95 *** (26.3)	23/94 *** (24.5)	0.770
TRM	35	16	19	
*Graft failure*		2	-	
*Infection*		6	6	
*GvHD*		3	7	
*Toxicity*		3	6	
*Other*		2	-	
Relapse/Progression	13	9	4	

* Number of evaluable HSCTs. ** Predefined conditions: high serological risk; haploidentical transplant; any HSCT subsequent to the first; acute GvHD; chronic GvHD; steroid dose > 2 mg/kg/day; second-line immunosuppressive drugs for the treatment of GvHD. *** Number of patients. Legend: non-malignant diagnosis = immunodeficiencies (*n* = 35, 31.5%), inherited bone marrow failure syndromes (*n* = 33, 29.8%), severe aplastic anemia (*n* = 26, 23.4%), thalassemia (*n* = 6, 5.4%), sickle cell disease (*n* = 5, 4.5%), autoinflammatory diseases (*n* = 3, 2.7%), metabolic diseases (*n* = 3, 2.7%); malignant diagnosis = acute lymphoblastic leukemia (*n* = 51 49.5%), acute myeloid leukemia (*n* = 34, 33%), juvenile myelomonocytic leukemia (*n* = 3, 2.9%), myelodysplastic syndrome (*n* = 7, 6.8%), lymphoma (*n* = 7, 6.8%), recurrent neuroblastoma (*n* = 1, 1%); AD = alternative donor; Haplo = haploidentical related donor; MRD = matched related donor; PT-Cy = post-transplant cyclophosphamyde; TRM = transplant-related mortality.

**Table 2 diagnostics-14-02461-t002:** Characteristics of the 69 HCTs complicated by CMV reactivation according to the execution of pre-emptive antiviral therapy.

		Pre-Emptive Antiviral Therapy		
Factors	Total, *n* = 69	Yes, *n* = 59 (85.5)	No, *n* = 10 (14.5)	*p*-Value
Sex, Males, *n* (%)	37 (53.6)	33 (55.9)	4 (40)	0.350
Age at HCT, years, median (1st–3rd quartiles)	8.5 (4.7–13.6)	7.9 (4.6–13.6)	10.4 (6.3–17.5)	0.200
Malignant diagnosis, *n* (%)	35 (50.7)	29 (49.1)	6 (60)	0.526
High serological risk, *n* (%)	21 (30.4)	20 (33.9)	1 (10)	0.263
Donor type, *n* (%)				0.082
Haplo	18 (26.1)	17 (28.8)	1 (10)	
AD	36 (52.2)	32 (54.2)	4 (40)	
MRD	15 (21.7)	10 (17)	5 (50)	
Source of stem cell graft, *n* (%)				0.248
Bone marrow	54 (78.3)	44 (74.6)	10 (100)	
Peripheral blood	13 (18.8)	13 (22)	0	
Cord blood	2 (2.9)	2 (3.4)	0	
Conditioning regimen, *n* (%)				1.000
MAC	41 (59.4)	35 (59.3)	6 (60)	
RIC	28 (40.6)	24 (40.7)	4 (40)	
Methylprednisolone > 2 mg/kg, *n* (%)	28 (40.6)	26 (44.1)	2 (20)	0.184
Second-line immunosuppressive therapy, *n* (%)	17 (24.6)	14 (23.7)	3 (30)	0.699
Acute GvHD, *n* (%)	32/65 * (49.2)	30/56 * (53.6)	2/9 * (22.2)	0.149
Chronic GvHD, *n* (%)	18/62 * (29.0)	18/54 * (33.3)	0/8 *	0.092
Presence of at least one predefined condition at risk for CMV **, *n* (%)	56 (81.2)	52 (88.1)	4 (40)	**0.002**
CMV-DNA copies, median (1st–3rd quartiles) at the beginning of the therapy	-	2200 (800–4500)	-	-
CMV-DNA copies, median (1st–3rd quartiles) at maximum monitoring value	5300 (2000–17050)	6600 (3150–19700)	1100 (300–1650)	**<0.001**
Days from HCT to reactivation, median (1st–3rd quartiles)	23 (15–43)	24 (15–43)	19.5 (15–40)	0.547
CMV reactivation after HCT subsequent to the first, *n* (%)	9 (13)	8 (13.6)	1 (190)	0.594
Deaths, *n* (%)	17/65 *** (26.1)	16/55 *** (29.1)	1/10 *** (10)	0.270

* Number of evaluable HSCTs. ** Predefined conditions: high serological risk; haploidentical transplant; any HSCT subsequent to the first; acute GvHD; chronic GvHD; steroid dose > 2 mg/kg/day; second-line immunosuppressive drugs for the treatment of GvHD. *** Number of patients. Legend: AD = alternative donor; Haplo = haploidentical related donor; MRD = matched related donor; MAC = myeloablative conditioning regimen; RIC = reduced-intensity conditioning regimen.

## Data Availability

The raw data supporting the conclusions of this article will be made available by the authors on request.

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
