# Peer review of "Monitoring and Management of Cytomegalovirus Reactivations After Allogeneic Hematopoietic Stem Cell Transplantation in Children: Experience from a Single Pediatric Center"

_diagnostics, 2024, doi:10.3390/diagnostics14212461_

Round 1
Reviewer 1 Report
Comments and Suggestions for Authors
The manuscript #diagnostics-3248323 titled “Monitoring and management of cytomegalovirus reactivations after allogeneic hematopoietic stem cell transplantation in children: a single pediatric centre experience” described the monitoring and management of CMV reactivation in pediatric stem cell recipients. I have the following comments,
1. Please provide the reference(s) for the cut-off values to initiate the pre-emptive therapy. Why was the cut-off value the same for high and low risk groups?
2. In the method section, the authors mentioned that the lower limit of CMV detection was 250 copies/mL and the upper limit was 107 copies/mL, please check.
3. In my opinion, the conversion factor for converting from copies/mL to IU/mL depends on the kits used. It is not necessarily equal to 1 IU/mL for 4 copies/mL.
4. For the results in table 1, some of the risks of CMV reactivation such as use of methylprednisolone, acute or chronic GvHD, did the authors confine the factors that happen only before CMV reactivation?
5. For the survival analysis, how long were the patients followed up? This should be clearly specified.
6. It was quite confusing. The first part of the results mentioned that there were 100 episodes of CMV reactivation while in the part of “pre-emptive therapy”, there were 69 episodes. In addition, the description in the “pre-emptive” part was quite confusing. Why did the authors separate the patients into having and not having pre-emptive therapy and analyzed the factors associated with this? In my opinion, it would be more interesting to analyze the outcomes after the treatment eg. graft failure, mortality, GvHD and explore whether having or not having pre-emptive therapy affecting these outcomes or not.
7. For the Kaplan-Meier curve, have the authors adjusted for potential factors associated with the survival?
8. How did the authors derive the flow diagram in figure 2 according to the results in the study?
9. The manuscript should be corrected by an English native speaker and the content should be re-organized.
Comments on the Quality of English LanguageThe manuscript should be corrected by an English native speaker and the content should be re-organized.
Author Response
Dear Sir,
Please find enclosed the revised version of the manuscript n. 3248323, entitled: “Monitoring and management of cytomegalovirus reactivations after allogeneic hematopoietic stem cell transplantation in children: experience from a single pediatric center”.
We thank the Reviewers for the positive feedback, we answered all the questions raised by the Reviewers and the main changes in the text are highlighted in yellow (Reviewer #1) and green (Reviewer#2) in the manuscript. Line numbers refer to the “Word” version of the document.
Reviewer 1.
- Please provide the reference(s) for the cut-off values to initiate the pre-emptive therapy. Why was the cut-off value the same for high and low risk groups?
The cut-off values that have been taken into consideration are listed in the introduction and refer to the bibliographic note 2. The cut-off values used in clinical practice, described for the period 2012-2019, are the result of literature data and of experience of the center.
In the Introduction, we reported the cut-off values considered in the adult population (ref n° 2) that differ in the high-risk vs low-risk group and in the presence/absence of CMV prophylaxis with letermovir. While in our clinical practice in children (and without any active CMV prophylaxis), we described a different approach based on the patients' risk assessment.
- In the method section, the authors mentioned that the lower limit of CMV detection was 250 copies/mL and the upper limit was 107 copies/mL, please check
We changed with 10^7 copies/mL (line 107-108)
- In my opinion, the conversion factor for converting from copies/mL to IU/mL depends on the kits used. It is not necessarily equal to 1 IU/mL for 4 copies/mL
Thank you for this comment. In this regard, we added a sentence that specifies this (line 51-54). Anyway, we would like to specify an example of conversion considering that the unit of measure suggested by WHO is IU/mL.
- For the results in table 1, some of the risks of CMV reactivation such as use of methylprednisolone, acute or chronic GvHD, did the authors confine the factors that happen only before CMV reactivation?
In Table 1 we compared factors between the group of HSCTs complicated by at least one CMV reactivation (n=100), independently from timing of reactivation, and the group of HSCTs without CMV reactivation.
- For the survival analysis, how long were the patients followed up? This should be clearly specified
Follow up was censored on July 2023 and patients were followed from HSCT to the last follow-up or death for a median of 3.1 [IQR 1.2-5.5] years (line 203-204) and a maximum value of 10 years.
- It was quite confusing. The first part of the results mentioned that there were 100 episodes of CMV reactivation while in the part of “pre-emptive therapy”, there were 69 episodes. In addition, the description in the “pre-emptive” part was quite confusing. Why did the authors separate the patients into having and not having pre-emptive therapy and analyzed the factors associated with this? In my opinion, it would be more interesting to analyze the outcomes after the treatment eg. graft failure, mortality, GvHD and explore whether having or not having pre-emptive therapy affecting these outcomes or not
Data on the use of pre-emptive therapy (based on viremia values) were only available in the period 2012-2019 (line 103-104) for 163 HSCTs and 69 (42.3%) (line 214-215) HSCTs were complicated by at least one CMV. During the entire study period 2012-2022, 214 allogeneic HSCTs were performed and 100 were complicated by at least one CMV (line 183).
We compared the two groups of patients, treated versus untreated, to analyze if there were any differences in characteristics and to retrospectively evaluate patient management. The outcome of untreated and treated patients in terms of frequency of acute, chronic GvHD and mortality was reported in Table 2.
- For the Kaplan-Meier curve, have the authors adjusted for potential factors associated with the survival?
We would like to underline that the main focus of this study was investigating the incidence of CMV reactivations after allogenic HSCT and analyzing the potential impact of recipient/donor-related transplant-related factors on CMV reactivation. Accordingly survival multivariable analysis was not performed, if the Reviewer agrees, we would prefer not focusing on this analysis.
- How did the authors derive the flow diagram in figure 2 according to the results in the study?
The figure lists the risk factors for CMV reactivation that emerged from the analysis of our population and those described in the literature on the subject. The specific reasons are described in the section “Discussion” from line 298 to line 314. The cut-off values, that are suggested in figure 2, are the result of what has been learned from the literature and from the retrospective analysis performed during the period between 2012-2019.
- The manuscript should be corrected by an English native speaker and the content should be re-organized
Corrections were made in the new version of the manuscript
Reviewer 2 Report
Comments and Suggestions for Authors
Dear Sir
I have carefully read the contribution by Ferrando G and collaborators:
Monitoring and management of cytomegalovirus reactivations afte allogenic hematopoietic stem cell transplantation in children: a single pediatric centre experience.
Manuscript ID: diagnostics-3248323
The authors report their experience, gained in a large Pediatric Hospital with teaching activities, in the management of the reactivation of CMV infections after allogeneic hematopoietic stem cell transplantation.
The work appears substantially well conceived.
The introduction adequately presents the assumptions and knowledge on the subject and is of adequate length.
The materials and methods section is adequate and explains in an exhaustive manner the methodological approach followed in the statistical processing of the results. Usually the IQR parameter (which is obtained by subtracting the value obtained for the first quartile from that obtained for the third quartile) is expressed as a single number and not as an interval between two extremes (could it be the minimum-maximum range?). This occurs repeatedly in the text.
The results section appears adequate. Personally, I would suggest changing the name of the "definitions" section with another term.
The sentence reported on lines 199-200 must be rewritten in a more understandable form.
The discussion section is clear, of adequate length. The suggested considerations are well supported by the data obtained.
The sentence included between lines 285-289 must be described in a more understandable form.
The bibliography is adequate. No self-citation by the authors is detected. The bibliographic entry number 18 must be formally adapted to the editorial standards of the journal.
Comments on the Quality of English LanguageA careful linguistic revision of the text and tables is recommended to check the spacing between words.
Author Response
Dear Sir,
Please find enclosed the revised version of the manuscript n. 3248323, entitled: “Monitoring and management of cytomegalovirus reactivations after allogeneic hematopoietic stem cell transplantation in children: experience from a single pediatric center”.
We thank the Reviewers for the positive feedback, we answered all the questions raised by the Reviewers and the main changes in the text are highlighted in yellow (Reviewer #1) and green (Reviewer#2) in the manuscript. Line numbers refer to the “Word” version of the document.
Reviewer 2.
The authors report their experience, gained in a large Pediatric Hospital with teaching activities, in the management of the reactivation of CMV infections after allogeneic hematopoietic stem cell transplantation.
The work appears substantially well conceived.
The introduction adequately presents the assumptions and knowledge on the subject and is of adequate length.
The materials and methods section is adequate and explains in an exhaustive manner the methodological approach followed in the statistical processing of the results. Usually the IQR parameter (which is obtained by subtracting the value obtained for the first quartile from that obtained for the third quartile) is expressed as a single number and not as an interval between two extremes (could it be the minimum-maximum range?). This occurs repeatedly in the text.
We thank the Reviewer for this suggestion, we changed the term “IQR” to “first and third quartiles” throughout the whole manuscript including tables.
The results section appears adequate. Personally, I would suggest changing the name of the "definitions" section with another term
We have deleted this term by including the section in the “study design”
The sentence reported on lines 199-200 must be rewritten in a more understandable form
The sentence was changed from line 200-202
The discussion section is clear, of adequate length. The suggested considerations are well supported by the data obtained.
The sentence included between lines 285-289 must be described in a more understandable form
The sentence was changed from line 288-292
The bibliography is adequate. No self-citation by the authors is detected. The bibliographic entry number 18 must be formally adapted to the editorial standards of the journal
The reference was changed from line 101-102 of the bibliography
Round 2
Reviewer 1 Report
Comments and Suggestions for Authors
The manuscript #diagnostics-3248323R1 titled “Monitoring and management of cytomegalovirus reactivations after allogeneic hematopoietic stem cell transplantation in children: a single pediatric centre experience” described the monitoring and management of CMV reactivation in pediatric stem cell recipients. I still have the following comments,
1. For the results in table 1, the authors may have to strict to the time sequences between some of the risks of CMV reactivation such as use of methylprednisolone, acute or chronic GvHD and the time of CMV reactivation. To define that these factors may be associated with CMV reactivation, these factors had to occur before CMV reactivation. did the authors confine the factors that happen only before CMV reactivation?
2. Regarding survival analysis, the authors may have to explore which factors could affect the survival and these factors should be used to adjust before concluding that the survival between CMV reactivation and no CMV reactivation groups were not different.
3.
Comments on the Quality of English LanguageSome grammatical correction is still needed.
Round 3
Reviewer 1 Report
Comments and Suggestions for Authors
The issues have been addressed.